# Human TLR8 induces inflammatory bone marrow erythromyeloblastic islands and anemia in SLE-prone mice

Naomi I Maria[1,2,*] ⬥, Julien Papoin[1,2,*] ⬥, Chirag Raparia[1,2] ⬥, Zeguo Sun[3], Rachel Josselsohn[1], Ailing Lu[1], Hani Katerji[4], Mahrukh M Syeda[5], David Polsky[5], Robert Paulson[6], Theodosia Kalfa[7] ⬥, Betsy J Barnes[1,2], Weijia Zhang[3], Lionel Blanc[1,2,†] ⬥, Anne Davidson[1,2,†] ⬥

Anemia commonly occurs in systemic lupus erythematosus, a disease characterized by innate immune activation by nucleic acids. Over-activation of cytoplasmic sensors by self-DNA or RNA can cause erythroid cell death, while sparing other hematopoietic cell lineages. Whereas chronic inflammation is involved in this mechanism, less is known about the impact of systemic lupus erythematosus on the BM erythropoietic niche. We discovered that expression of the endosomal ssRNA sensor human TLR8 induces fatal anemia in Sle1.Yaa lupus mice. We observed that anemia was associated with a decrease in erythromyeloblastic islands and a block in differentiation at the CFU-E to proerythroblast transition in the BM. Single-cell RNAseq analyses of isolated BM erythromyeloblastic islands from human TLR8-expressing mice revealed that genes associated with essential central macrophage functions including adhesion and provision of nutrients were down-regulated. Although compensatory stress erythropoiesis occurred in the spleen, red blood cell half-life decreased because of hemophagocytosis. These data implicate the endosomal RNA sensor TLR8 as an additional innate receptor whose overactivation causes acquired failure of erythropoiesis via myeloid cell dysregulation.

## Introduction

Anemia is the most common hematologic abnormality in patients with systemic lupus erythematosus (SLE) and often accompanies disease flare (Velo-García et al, 2016; Anderson et al, 2018; Akilesh et al, 2019). The commonest cause of anemia in SLE is ineffective erythropoiesis mediated by inflammatory cytokines or autoantibodies (Liu et al, 1995; Velo-García et al, 2016). An increase in erythroid cell death within the hematopoietic niche has been identified in the BM of active SLE patients (Feng et al, 1991; Zhuang et al, 2014) and reticulocytes with incomplete mitophagy can be found in the circulation (Ney, 2011; Caielli et al, 2021). Damage to the BM niche is reproduced in mice by administering pristane and requires both TLR7 and TNF (Zhuang et al, 2014). By contrast, constitutive failure to clear intracellular self-DNA causes defective erythropoiesis in the fetal liver in an IFNAR-dependent manner (Kawane et al, 2001). Additional syndromes induced by altered RNA editing in erythroblasts (Liddicoat et al, 2016; Nakahama & Kawahara, 2020) are rescued by deficiency of RNaseL or the RNA sensor MDA5 (Liddicoat et al, 2015; Daou et al, 2020). These data indicate that erythropoiesis is sensitive to excessive immune stimulation by both self-DNA and self-RNA leading to erythroid cell death, while sparing other hematopoietic cell lineages.

The endosomal receptors TLR7 and TLR8 recognize ssRNA and RNA-derived nucleotides and overexpression of either receptor in mice induces autoantibodies and systemic inflammation (Pisitkun et al, 2006; Guiducci et al, 2013). These TLRs differ in their cellular distribution, the RNA sequences they recognize (Greulich et al, 2019; Ostendorf et al, 2020), and the cytokine responses they induce (Greulich et al, 2019; Vierbuchen et al, 2019; Bender et al, 2020; Ostendorf et al, 2020). Overexpression of TLR7 in normal mice causes a lupus-like syndrome that includes the development of moderate anemia and thrombocytopenia (Deane et al, 2007; Akilesh et al, 2019) with normal BM. Furthermore, patients with mosaic TLR8 gain-of-function present immunodeficiency, inflammation, anemia, and BM failure (Aluri et al, 2021).

The role of TLR8 is difficult to study in mice because a five-amino acid deletion in mouse Tlr8 attenuates its RNA-binding capacity (Heil et al, 2004). Here, we analyzed the role of human TLR8 in a

[1]Institute of Molecular Medicine, Feinstein Institutes for Medical Research, Manhasset, NY, USA  [2]Donald and Barbara Zucker School of Medicine at Northwell Health, Hempstead, NY, USA  [3]Department of Medicine, Mount Sinai Medical Center, New York, NY, USA  [4]Department of Pathology, University of Rochester, Rochester, NY, USA  [5]The Ronald O. Perelman Department of Dermatology, New York University Grossman School of Medicine, New York, NY, USA  [6]Department of Veterinary and Biomedical Sciences, Penn State College of Agricultural Sciences, University Park, PA, USA  [7]Cincinnati Children's Hospital Medical Center, Cincinnati, OH, USA

Correspondence: adavidson1@northwell.edu
*Naomi I Maria and Julien Papoin are co-first authors
†Lionel Blanc and Anne Davidson are co-senior authors

mouse lupus model expressing a human TLR8 BAC transgene (huTLR8tg) that confers low-level huTLR8 expression, insufficient to cause spontaneous inflammation in non-autoimmune mice (Guiducci et al, 2013). We bred the transgene into Sle1.Yaa mice in which the *Sle1* locus causes loss of B cell tolerance without clinical disease (Mohan et al, 1998) and the addition of the Y accelerator (*Yaa*) locus, a translocation of a portion of the X chromosome onto the Y chromosome, confers an extra copy of *Tlr7* and induces pathogenic autoantibodies and lupus nephritis in Sle1.Yaa males (Morel et al, 2000).

We observed that huTLR8 induces fatal anemia in adult male Sle1.Yaa mice. The anemia in huTLR8tg.Sle1.Yaa mice is caused by a block in differentiation from CFU-E to proerythroblasts (ProEB) in the BM and is associated with an altered phenotype of the central macrophages in the erythroid niche. Traditionally known as erythroblastic islands, we recently observed that granulopoiesis also occurs within these anatomical structures and renamed them erythromyeloblastic islands (EMBIs [Romano et al, 2022]). Compensatory stress erythropoiesis in the spleen results in the expansion of several subsets of inflammatory hemophagocytes (Akilesh et al, 2019), and fatal anemia is associated with a decrease in RBC half-life, suggesting that the combination of BM failure and RBC phagocytosis eventually exceeds the capability of stress erythropoiesis to replace the RBC mass. Inducible anemia caused by an excess burden of endosomal TLR ligands could help explain the anemia found in patients with active SLE.

# Results

## huTLR8 is functional in Clone 8 mice

Quantitative RT–PCR analyses from the spleens of Clone 8 mice (Guiducci et al, 2013) showed increased expression of huTLR8 mRNA (Fig S1A). HuTLR8 protein expression was detected by immunohistochemistry in the spleens of huTLR8tg mice (Fig S1B). The TLR4 agonist LPS and the TLR7 agonist CL264 induced TNF in BM macrophages of both C57BL/6 and huTLR8tg mice, but the TLR8 agonist TL-506 induced TNF only in huTLR8tg BM macrophages (Fig S1C). Furthermore, the TLR7/8 agonists CL075 and TL8-506 induced glycolysis in TLR7-deficient BM macrophages only in the presence of the huTLR8 transgene (Fig S1D and E). As previously reported (Guiducci et al, 2013), the huTLR8 transgene was expressed predominantly in myeloid cells of Clone 8 mice (Fig S1F).

## huTLR8tg.Sle1.Yaa males die of severe anemia

Having demonstrated that huTLR8 is functional in mice, we bred Clone 8 mice to the Sle1.Yaa strain (Morel et al, 2000). Sle1.Yaa mice die of nephritis by 9–12 mo of age (Morel et al, 2000). Homozygous huTLR8tg.Sle1.Yaa males displayed earlier mortality (Fig 1A); this was not due to accelerated renal disease because none had developed proteinuria, and histologic examination of kidneys from moribund mice did not show renal inflammation (Fig S2A–E). Furthermore, autoantibody titers did not differ between huTLR8tg. Sle1.Yaa mice and Sle1.Yaa controls (Fig S2F–H), nor were the number of B and T cell subsets per spleen different (Fig S3B–H),

despite increased spleen weight in huTLR8tg.Sle1.Yaa mice (Fig S3A). By contrast, CD11b⁺ myeloid cells were increased in huTLR8tg.Sle1.Yaa spleens compared with Sle1.Yaa controls (Fig S3I and J). Complete blood counts revealed that homozygous huTLR8tg.Sle1.Yaa mice developed severe anemia (Hb ≤ 9 g/dl) and death occurred when the Hb level dropped below 6 g/dl. Severe anemia was absent in age-matched Sle1.Yaa mice, homozygous huTLR8tg.C57BL/6.Yaa (B6.Yaa tg) mice or homozygous female huTLR8tg.Sle1 mice (Fig 1B and Table S1). Injection of the TLR8 ligand TL8-506 in a dose sufficient to cause rapid onset of nephritis in Sle1.Yaa mice induced reticulocytosis and anemia in only 1/10 huTLR8tg.B6.Yaa mice (Fig S3K and L). Together, these findings indicate that in addition to the huTLR8 transgene, both the *Sle1* and *Yaa* loci are required for the fatal anemia phenotype. Longitudinal analysis of 12 huTLR8tg.Sle1.Yaa mice that developed severe anemia showed that Hb was normal before 10 wk of age and that anemia subsequently developed over a variable period (Fig 1C). A compensatory increase in serum levels of erythropoietin (Fig 1D) along with reticulocytosis (Fig 1E) occurred in anemic mice. Mild thrombocytopenia developed in both transgenic and non-transgenic mice compared with B6.Yaa controls probably because of the *Yaa* translocation (Fig 1E and Table S1).

## huTLR8tg.Sle1.Yaa mice develop BM failure and stress erythropoiesis

We next investigated erythropoiesis in homozygous huTLR8tg.Sle1.Yaa mice. The frequency of erythroid precursors was decreased in the BMs of mice with severe anemia but not in non-anemic mice or non-transgenic controls (Figs 2A and B and S4B and E). However, the progression from pro- to orthochromatic erythroblasts was unaffected by the presence of the transgene (Fig 2C), showing that terminal erythropoiesis in the BM proceeds normally.

In anemic mice, compensatory erythropoiesis occurs in the spleen, through stress erythropoiesis (Paulson et al, 2020). Accordingly, we observed splenomegaly (Fig S3A), expansion of the red pulp (Figs 2D and E and S4I compared with Fig S4F–H), and increased terminal erythroid differentiation in the spleens of anemic huTLR8tg.Sle1.Yaa mice (Figs 2F and G and S4A, C, and D). Together, these data suggest that stress erythropoiesis cannot compensate for the failed BM erythropoiesis observed in huTLR8tg.Sle1.Yaa mice.

## Decreased survival and increased phagocytosis of erythrocytes in huTLR8tg.Sle1.Yaa mice

In addition to failure of BM erythropoiesis, we hypothesized that decreased red cell (RBC) survival contributes to fatal anemia in huTLR8g mice. Using in vivo biotin labeling, we observed that RBC survival was decreased in huTLR8tg.Sle1.Yaa mice, but only after anemia had developed (Fig 3A, dark red versus green lines). In addition to stress erythropoiesis, myeloid cells were expanded in the spleens of transgenic mice (Fig S3I and J), suggesting that hemophagocytosis contributes to decreased RBC survival. To test this hypothesis, we performed flow cytometry using intracellular Ter119 staining to detect hemophagocytic cells (Table S2). We detected three Ter119⁺subsets in anemic huTLR8tg.Sle1.Yaa mice but

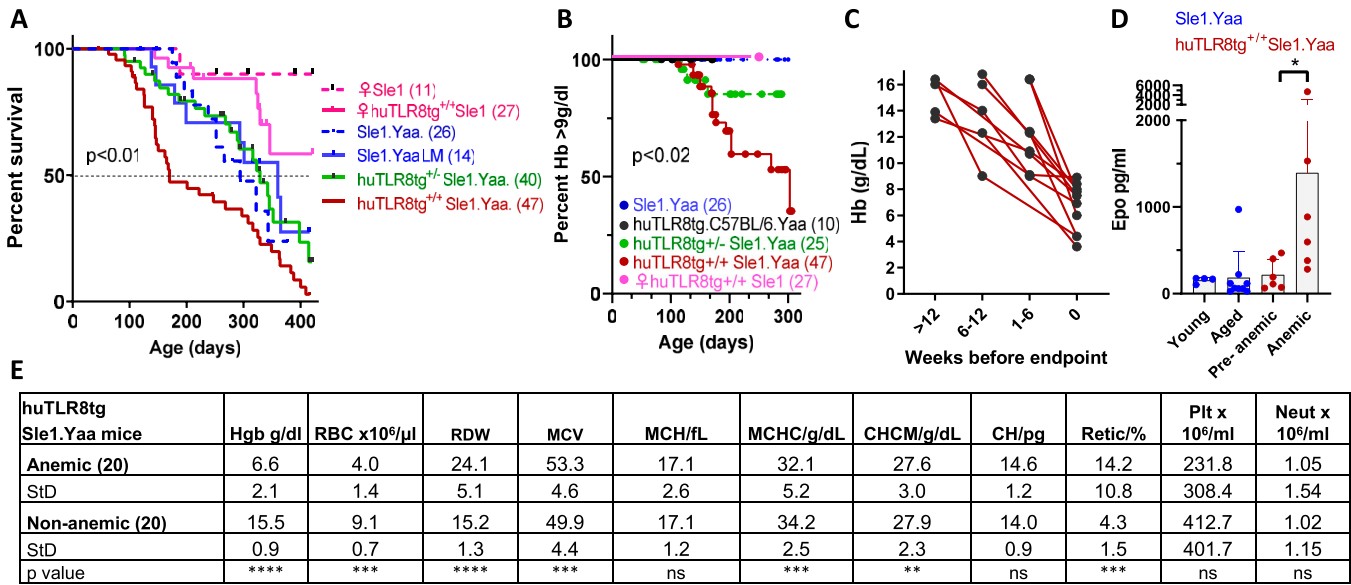

**Figure 1. huTLR8tg.Sle1.Yaa mice develop severe anemia.**
**(A)** Survival of mice of the indicated sex and genotype LM = littermate control; *P*-value calculated using Log rank Mantel-Cox test (*P* < 0.01, huTLR8[+/+]Sle1.Yaa versus other male groups). **(B)** Development of severe anemia (Hb ≤ 9 g/dl) in mice of the indicated genotype; *P*-value calculated using Log rank Mantel-Cox test (*P* < 0.02, huTLR8[+/+].Sle1.Yaa versus other male groups). **(C)** Time frame for the development of severe anemia in 12 individual huTLR8tg.Sle1.Yaa mice observed longitudinally. Time 0 indicates the Day when Hb reaches ≤ 9 g/dl. **(D)** Serum levels of erythropoietin in six huTLR8tg.Sle1.Yaa mice in which serum samples were obtained longitudinally before (Hb ≥ 14g/dl) and after severe anemia onset. Young and aged Sle1.Yaa controls are shown for comparison. *P*-value calculated using Wilcoxon matched pairs test. *P* < 0.05. Each data point represents a single mouse. **(E)** Blood counts in severely anemic huTLR8tg.Sle1.Yaa mice and age-matched non-anemic huTLR8tg.Sle1.Yaa controls. HGB, hemoglobin; RBC, red blood cells; RDW, red cell distribution width; MCV, mean corpuscular volume; MCH, mean corpuscular hemoglobin; MCHC, mean corpuscular hemoglobin concentration; CHCM, cellular hemoglobin concentration mean; CH, cellular hemoglobin; Retic, reticulocytes; Plt, platelets; Neut, neutrophils. *P*-values calculated using Mann–Whitney nonparametric *t* test.

not in young huTLR8tg.Sle1.Yaa mice or age-matched Sle1.Yaa controls. These included F4/80[hi]/VCAM[hi] RPMs, CD31[+]/F4/80[int] pre-red pulp macrophages, and a previously defined subset of CD31[+]/F4/80[−] inflammatory hemophagocytes (Akilesh et al, 2019 - Figs 3B–E, S5A, and S6A and B). These findings were further supported by in vivo phagocytosis assays using CFSE-labeled RBCs (Bennett et al, 2019). 24 h after transfusion, CFSE-labeled RBCs were detected in all three subsets of anemic mice, but not in neutrophils, F4/80[−]/CD11c[+] DCs or Ly6C[hi] macrophages (Figs 3F and G and S5B). Expansion of hemophagocytic macrophages in transgenic mice was further confirmed by histologic examination of spleens (Fig 3H–K). Inflammation can induce changes in RBCs that make them more prone to hemophagocytosis (Caielli et al, 2021; Lam et al, 2021). To investigate this possibility, we transferred biotinylated RBCs from anemic hu.TLR8tg.Sle1.Yaa to young Sle1.Yaa mice. RBCs from anemic mice had a normal half-life when they were transferred into young Sle1.Yaa mice, showing that an intrinsic RBC defect did not account for the hemophagocytosis. Conversely, CFSE-labeled cells from young Sle1.Yaa controls had a decrease in half-life when they were transferred to anemic huTLR8tg.Sle1.Yaa mice (Fig 3L).

### EMBIs are abnormal in anemic huTLR8tg.Sle1.Yaa mice

Bones of anemic mice appeared thicker and paler than those of their non-anemic counterparts and the marrow was filled with trabecular bone (Fig 4A). Serum levels of osteocalcin were higher in anemic compared with pre-anemic sera from the same mice with a decrease in TRAP (Fig 4B and C). Erythropoiesis occurs in specialized niches in which a central macrophage provides nutrients to differentiating erythroid precursors and safely ingests the pyrenocyte (Manwani & Bieker, 2008; Hom et al, 2015; Mukherjee et al, 2021). We recently demonstrated these BM niches also foster granulopoiesis, naming them EMBIs. To further characterize the niche, we examined EMBIs from the BM and spleens of huTLR8tg.Sle1.Yaa and WT Sle1.Yaa mice using Amnis ImageStream (Fig S7A). BMs from anemic huTLR8tg.Sle1.Yaa mice contained a lower frequency of EMBIs than those of their WT counterparts, including non-anemic huTLR8tg.-Sle1.Yaa mice, age-matched Sle1.Yaa mice or B6.Yaa controls, whereas spleens contained increased numbers because of spleen enlargement (Fig 4D). No differences were observed in the proportions of myeloid and erythroid cells within the EMBIs of huTLR8tg.Sle1.Yaa mice compared with B6.Yaa and Sle1.Yaa controls (Fig 4E).

We next performed flow cytometry of erythroid progenitors using the gating strategy shown in Fig S7B–D (Tusi et al, 2018). No differences in the frequency of cKit[+]Sca1[−] or CD55[+] progenitors were observed between Sle1.Yaa and huTLR8tg.Sle1.Yaa mice (Fig S7E and F). Similarly, there was no difference in the percent of BFU-E and CFU-E or the ratio of CFU-E to BFU-E between WT and transgenic mice (Fig 4F–H). However, we observed a block in differentiation at the CFU-E to ProEB transition in the huTLR8tg.Sle1.Yaa mice and this block was associated with reticulocytosis, indicative of stress erythropoiesis (Fig 4I–M). Flow cytometry of myeloid progenitors (Muench et al, 2020) showed no differences between Sle1.Yaa and

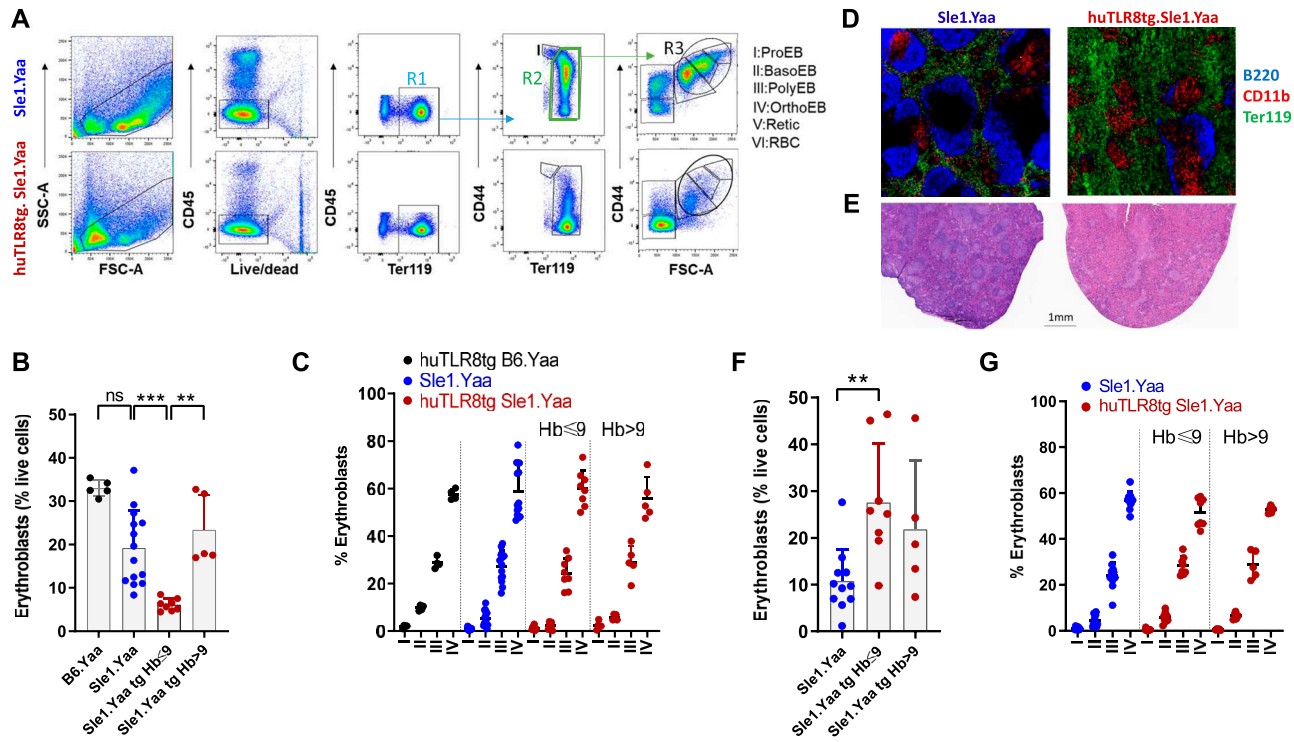

**Figure 2. huTLR8tg Sle1.Yaa mice develop stress erythropoiesis to compensate for impaired terminal differentiation in the BM.**
**(A)** Gating strategy for the analysis of terminal erythroid differentiation in the non-lysed BM of Sle1.Yaa (upper panels) or huTLR8tgSle1.Yaa (lower panels) based on CD44/Ter119 and FSC as described in Liu et al (2013) enabling the identification of six populations (from I to VI). ProEB, proerythroblast; BasoEB, basophilic erythroblast; PolyEB, polychromatophilic erythroblast; orthoEB, orthochromatic erythroblast; retic, reticulocyte; RBC, red blood cell. **(B)** Percent erythroblasts based on the amount of live cells (R1). **(A, C)** Percent of each subpopulation of erythroblast from I to IV based on the gating strategy established in (A). **(D)** Representative immunohistochemistry of spleens from age-matched Sle1.Yaa (left panel) and anemic huTLR8tg.Sle1.Yaa (right panel) mice showing the expansion of Ter119 cells and disorganization of follicles in huTLR8tg.Sle1.Yaa mice (10X magnification). **(E)** H & E staining of spleens from age-matched Sle1.Yaa (left panel) and huTLR8tg Sle1.Yaa (right panel) shows expansion of the red pulp in the huTLR8tg Sle1.Yaa mice and disorganization of the follicular architecture. **(D, E)** Representative of at least three mice per strain. **(F)** Percent erythroblasts based on the amount of live cells in the spleen. **(A, G)** Percent of each erythroblast subpopulation in the spleen based on the gating strategy established in (A). **(B, C, F, G)** Kruskal–Wallis ANOVA followed by Dunn's multiple comparisons test \*\*P < 0.01; \*\*\*P < 0.001. Each data point represents a single mouse.

huTLR8tg.Sle1.Yaa mice (Fig S8A–C), highlighting specificity towards the erythroid compartment.

To clarify the contribution of EMBIs to the ProEB defect, we performed scRNAseq analyses of BM and spleen EMBIs isolated from age-matched Sle1.Yaa and severely anemic huTLR8Tg.Sle1.Yaa mice (Fig S9A). We analyzed 19,309 BM EMBI cells and 4,794 spleen EMBI cells that formed eight erythrocyte, seven granulocyte, seven macrophage, and two progenitor clusters (Fig 5A–C and Table S3). When the eight clusters of erythroid cells were considered together, we observed the preservation of E0 and E1 clusters in the BMs of anemic huTLR8tg.Sle1.Yaa mice but a decrease in the frequency of clusters E2–5, in particular, E2- defined as ProEB cluster (Fig 5A upper panels, Fig 5D and E).

The erythroid clusters could be assigned to sequential stages of erythroid cell development using gene expression (Fig S10A) and trajectory pseudotime analysis (Qiu et al, 2017) (Monocle) (Figs S10B and C and S11). We identified cluster E1 as CFU-Es based on the expression of *Gata-1* and *Klf1* but low levels of *Slc4a1, Glycophorin A,* and *Hba* (Figs 5B and S11). Differential gene expression of cluster E1 cells from BM EMBIs showed down-regulation of *Ki67, Slc4a1,* and *Aqp1* in huTLR8tg.Sle1.Yaa mice compared with Sle1.Yaa BMs. In

addition, there was a decrease in components of the electron transport chain (Figs 5F and S11 and Table S4). Gene enrichment analysis confirmed these findings, identifying a decrease in pathways involved in cell replication, cell cycle, and oxidative phosphorylation (Figs 5G and S12A). Up-regulated genes included genes involved in glycolysis (*Ldha* and *Aldoa*) (Figs 5F and S11). These abnormalities were not present in cluster E1 cells from spleens of anemic huTLR8tg.Sle1.Yaa mice in which there was robust stress erythropoiesis (Fig S11 and Table S4).

### Abnormal central macrophage function in anemic huTLR8tg.Sle1.Yaa mice

We subdivided the seven clusters of EMBI macrophages into two major groups based on their gene signatures (Figs 5B and C and S13A): the first contained clusters Mφ0 and Mφ1 and the second contained clusters Mφ2–4. An additional small cluster (Mφ5) had a mixed phenotype. Cluster Mφ6, found in Sle1.Yaa BM, expressed markers of pDCs and was not further considered as a macrophage subset. The frequency of each macrophage cluster within the BM and spleen EMBIs was different depending on the macrophage

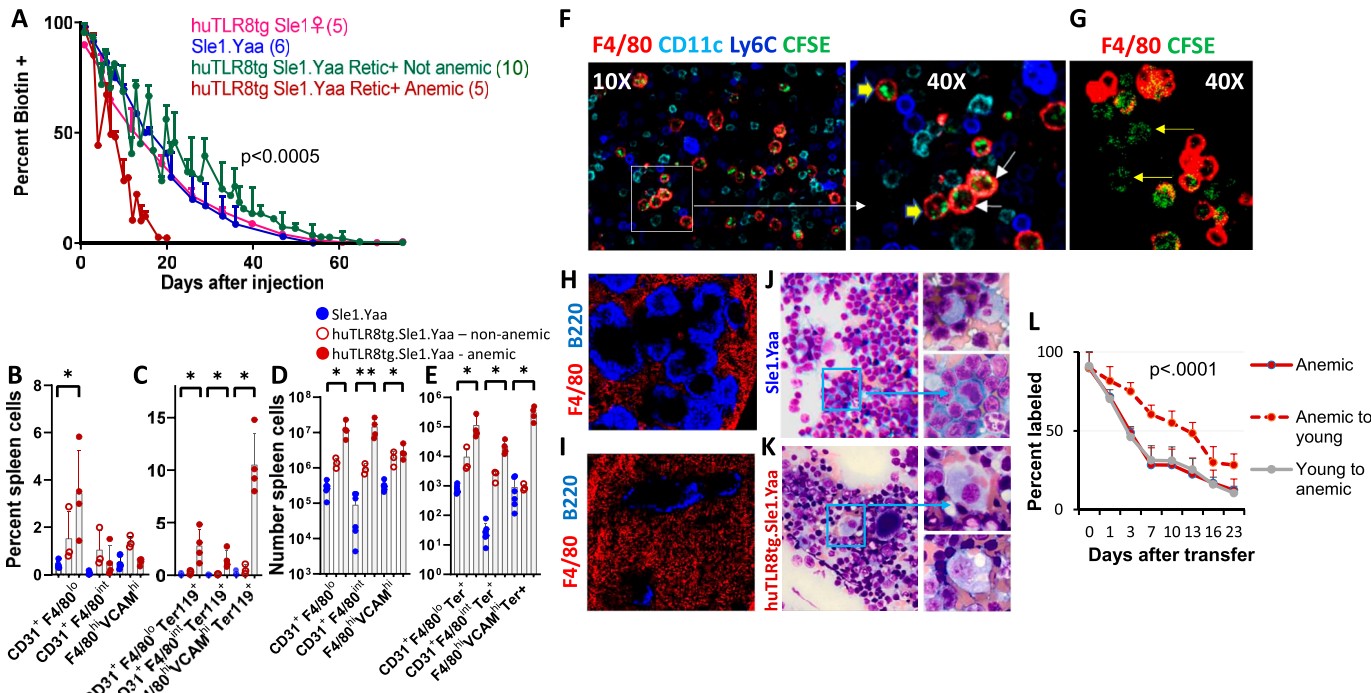

**Figure 3. Hemophagocytosis is a late manifestation of anemia in huTLR8tg Sle1.Yaa mice.**
**(A)** RBC survival was assessed by in vivo biotin labeling followed by sequential flow cytometry of peripheral blood RBCs. *P*-value calculated using Log rank Mantel-Cox test (huTLR8tg.Sle1.Yaa versus other groups). **(B, C, D, E)** Flow cytometry of myeloid cells cell subsets from anemic huTLR8tg.Sle1.Yaa mice (red) and age-matched Sle1.Yaa controls (blue) gated as shown in Figs S5 and S6 shows the percent of total (B) and hemophagocytic intracellular Ter119[*] (C) CD31[hi] (F4/80[lo] or [int]) and F4/80[hi]/VCAM[hi] RPMs and the number of each subset per spleen (D, E). Kruskal–Wallis ANOVA followed by Dunn's multiple comparisons test of all groups *P < 0.05. **(F)** Immunohistochemistry of cytospun spleen cells from a severely anemic huTLR8tg.Sle1.Yaa mouse 24 h after transfer of CFSE-labeled RBCs shows phagocytosis by F4/80[hi] (white arrows) and F4/80[int] (yellow arrowheads) macrophages but not by F4/80[−]/CD11c[hi] DCs or Ly6C[hi] monocytes. **(G)** Immunohistochemistry of cytospun spleen cells from a severely anemic huTLR8tg.Sle1.Yaa mouse 24 h after transfer of CFSE-labeled RBCs shows additional phagocytic F4/80[−] cells. **(H, I)** Expansion of F4/80[hi] cells in the spleen red pulp of an anemic huTLR8tg.Sle1.Yaa mouse (I) compared with an age-matched Sle1.Yaa control (H). **(J, K)** Touch preps of spleens show myeloid cell expansion and hemophagocytosis by macrophages from an anemic huTLR8tg.Sle1.Yaa mouse (K), compared with the absence of hemophagocytosis by macrophages from an age-matched Sle1.Yaa control (J). **(F, G, H, I, J, K)** Figures are representative of at least three mice per group. **(L)** RBCS from anemic huTLR8tg.Sle1.Yaa mice were biotin labeled in vivo and transferred to young (<8-wk-old) Sle1.Yaa mice then replaced with CFSE-labeled RBCs from the young mice. Biotin-labeled and CFSE-labeled RBCs were analyzed in the same mice (4 per group) and data are shown as percent of labeled cells (corrected to Day 0 values). Biotin marks the cells from the anemic donors and CFSE marks the cells from the young donors. *P*-value calculated using two-way ANOVA.

cluster and on the genotype (Fig 6A). Clusters Mφ0 and Mφ1 expressed markers of BM-derived monocytes, whereas clusters Mφ2–4 expressed markers of resident macrophages (Figs 6B and S13A and Table S3). Trajectory pseudotime analysis was used to analyze the cell fates of the six macrophage clusters. The cells diverged along two alternate paths from a central node (Node 4) towards either Mφ0–Mφ1 (Fate 2: BM monocytes) or Mφ2–Mφ5 (Fate 1: resident macrophages – Figs 6C and S13B and C).

Central macrophages express canonical markers *Adgre1* (F4/80), *Hmox1*, *Itgav*, *Mertk*, *Slc40a1*, and *Vcam1* (Toda et al, 2014; Seu et al, 2017; Sukhbaatar & Weichhart, 2018; Li et al, 2019). These markers, and *DNAse2a* (Li et al, 2019) and *Gdf15* (Hao et al, 2019) were preferentially expressed by cluster Mφ2 from Sle1.Yaa BM (Fig 6B). Transcription factors previously associated with EMBI macrophages, *Nr1h3*, *SpiC* and *Maf* (Kusakabe et al, 2011; Haldar et al, 2014; Li et al, 2019) were found only in clusters Mφ2 and Mφ4 (Table S3). Differential gene expression analysis further indicated that there was a decrease in expression of adhesion molecules *Vcam1* and *Itgav* and phagocytic receptors *Mertk*, *Axl*, *Mrc1*, and *Siglec1* in cluster Mφ2 macrophages from anemic huTLR8tg.Sle1.Yaa mice

compared with Sle1.Yaa controls. Other down-regulated genes included *Slc40a1*, *Fosb*, and *Jun*, genes that regulate Mφ2 macrophage polarization (Roy et al, 2015), *Igf1*, *IL18*, and *Vegfb*, growth factors needed for erythroblast maturation (Drogat et al, 2010; Kadri et al, 2015; Li et al, 2020) and *Egr1* that negatively regulates macrophage inflammatory genes (Trizzino et al, 2021). Up-regulated genes included multiple ribosomal proteins, several IFN-induced genes such as *Ifitm2* and *Plac8,* and pro-inflammatory genes including *Saa3* and *Pycard* (Figs 6D and S13C and Table S5). Additional differences within huTLR8tg.Sle1.Yaa mice were present between BM and spleen Mφ2 cells including in *Gdf15* and *SpiC* that are known to be expressed at high levels in spleen central macrophages in other models of stress erythropoiesis (Bennett et al, 2019) (Fig S13D). Overall, these findings indicate that cluster Mφ2 macrophages polarize to an altered phenotype in huTLR8tg.Sle1.Yaa mice with down-regulation of phagocytic receptors, growth factors, and adhesion molecules that are needed for central macrophage function. Gene set enrichment analysis confirmed these findings (Figs 6E and S12B). Further inspection of the pseudotime trajectory analysis indicated that

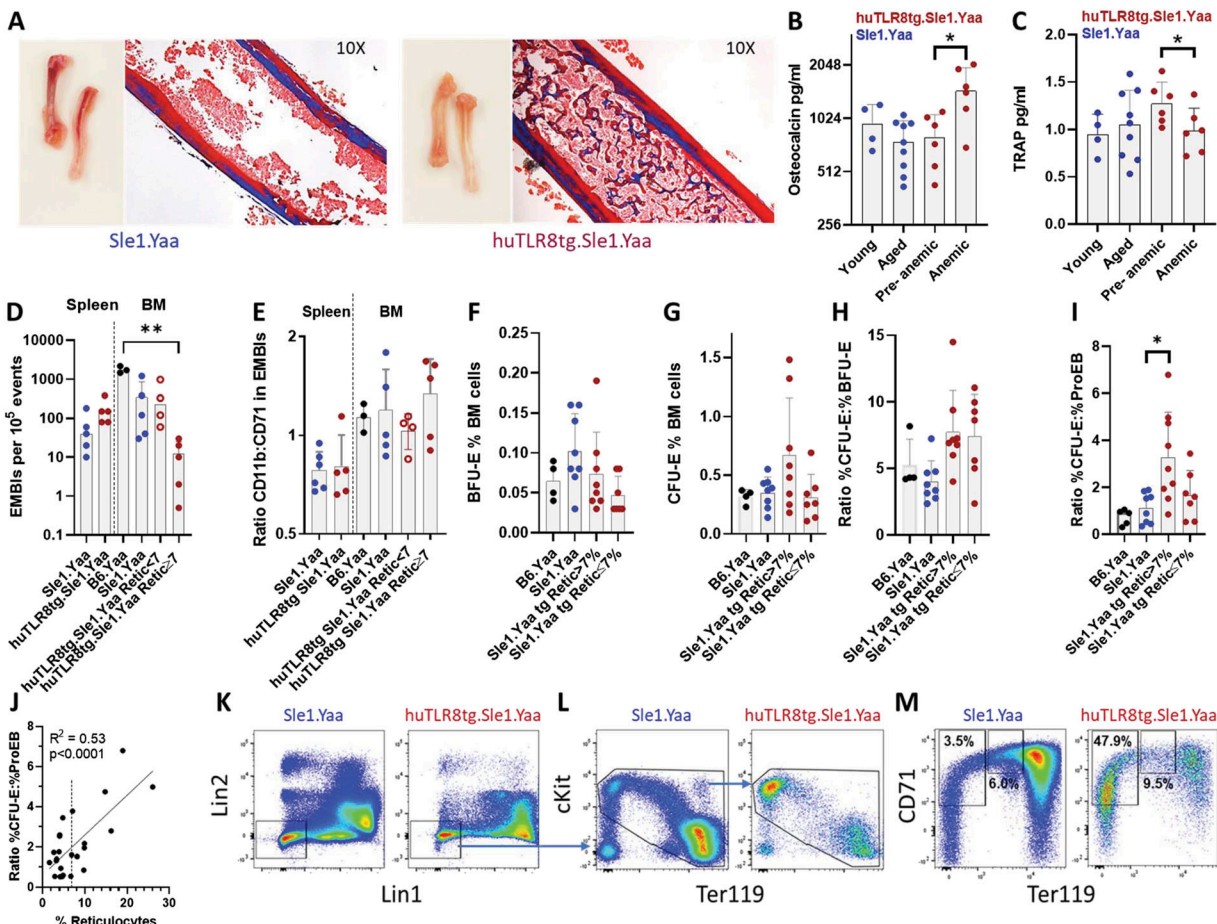

**Figure 4. Block in erythropoiesis at the CFU-E to proerythroblast transition.**
**(A)** Trichrome staining in huTLR8tg.Sle1.Yaa compared with age-matched Sle1.Yaa control (representative of three mice per strain). **(B, C)** Levels of osteocalcin (B) and tartrate-resistant acid phosphatase activity (C) in paired sera from huTLR8tg.Sle1.Yaa mice with unmatched sera from young and aged Sle1.Yaa mice for comparison—Wilcoxon matched pairs analysis for transgenic mice, *$P < 0.05$. Each data point represents a single mouse. **(D)** Frequency of EMBIs in BM spleen recorded by ImageStream analysis. **(E)** Ratio of CD71+ erythroblasts to CD11b+ myeloid cells in EMBIs (representative ImageStream images shown in Fig S5). **(F, G, H)** Flow cytometry analyses of BM erythroid progenitors show no change in the percent of BFU-E (F), CFU-E (G) or ratio of CFU-E:BFU-E (H) in huTLR8tg.Sle1.Yaa mice compared with age-matched Sle1.Yaa controls regardless of the reticulocyte count. **(I, J)** Increased ratio of CFU-E:ProEB in BMs of huTLR8tg.Sle1.Yaa mice (I) that correlates with reticulocyte count (J). **(D, E, F, G, H, I)** Kruskal–Wallis ANOVA followed by Dunn's multiple comparisons test of all groups *$P < 0.05$; **$P < 0.01$. **(J)** Simple linear regression. Each data point represents a single mouse. **(K, L, M)** Representative flow cytometry of BM cells gated for live cells as in Fig S5B and then as Lin− (K), cKit+, and Ter119+ (L) and then for CD71 and Ter119 (M). CFU-E is CD71+/Ter119lo and ProEB is CD71hi/Ter119int.

Mφ2–4 cells had two alternative fates diverging from Node 3; towards either VCAMhi (Fate 3: Sle1.Yaa BMs and huTLR8tg.Sle1.Yaa spleens) or VCAMlo (Fate 4: BMs of anemic huTLR8tg Sle1.Yaa mice) (Fig 6F and G).

To validate our scRNAseq data, we performed flow cytometry analyses of BM macrophages from age-matched huTLR8tg.Sle1.Yaa and Sle1.Yaa and B6.Yaa controls (Fig 7). F4/80hi BM macrophages from B6.Yaa and Sle1.Yaa mice expressed CD169, MERTK, and VCAM1, phenotypes consistent with cluster Mφ2 (Fig 7A, Gate R3) or were positive for CCR2 and Ly6C, phenotypes consistent with clusters Mφ0 and Mφ1 (Fig 7A, Gate R2). A smaller subpopulation expressed FcγR4 but not VCAM1 or CCR2 (Fig 7A, Gate R1). By contrast, F4/80hi BM macrophages from anemic huTLR8tg.Sle1.Yaa mice had decreased the expression of CD169 and VCAM1 and a partial loss of MERTK expression with an intermediate phenotype of non-anemic mice (Figs 7B–E and S14A). Findings were confirmed by ImageStream

analysis of EMBIs stained for F4/80, VCAM1, MERTK, and CD169 (Fig 7F). Spleen F4/80hi macrophages maintained the expression of both MERTK and VCAM1 consistent with the scRNAseq data (Fig S14B and C).

### Altered EMBI formation in huTLR8tg.Sle1.Yaa mice

To assess the capacity of EMBIs from huTLR8tg.Sle1.Yaa and Sle1.Yaa to form ex vivo, we performed reassociation experiments using either whole BM or isolated EMBIs (Figs S9B and S15A). We observed that the in vitro reassociation frequency was significantly lower in the severely anemic huTLR8tg.Sle1.Yaa mice than in Sle1.Yaa mice (Figs 7G and S15B), with a decrease in the expression of VCAM1, MERTK, and CD169 on reassociated EMBIs (Fig S15C). These findings support our contention that an acquired defect of the BM EMBIs in huTLR8tg.Sle1.Yaa mice results in decreased erythropoiesis.

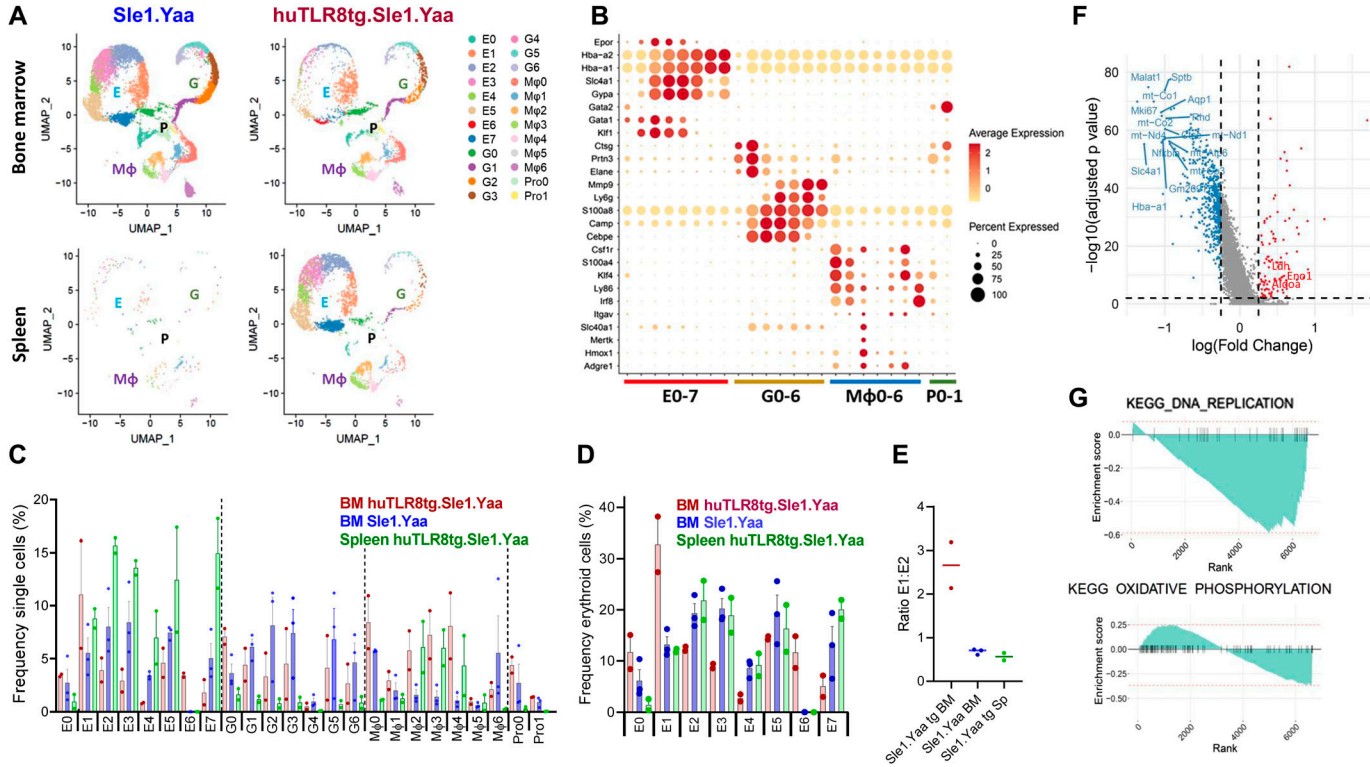

**Figure 5. scRNAseq reveals diverse macrophage subpopulations within EMBIs.**
**(A)** UMaps show the clusters of progenitor cells (P), erythroid cells (E), granulocytes (G), and macrophages (Mφ) in the indicated organs from Sle1.Yaa (left panels) and huTLR8tg.Sle1.Yaa (right panels) mice. **(B)** Bubble plot shows genes that differentiate the cell lineages. **(C)** Frequency of each cluster among the total cell population. **(D)** Frequency of cells in each cluster among cells belonging to the erythroid lineage. **(E)** Ratio of percent cells in Cluster E1 relative to E2. **(F)** Volcano plot shows differentially expressed genes in Cluster E1 (CFU-E) from huTLR8tg.Sle1.Yaa BM compared with Sle1.Yaa BM. **(G)** Pathways down-regulated in Cluster E1 (CFU-E) from huTLR8tg.Sle1.Yaa BM compared with Sle1.Yaa BM, identified by KEGG analysis (see Fig S12A).

To address whether the defect in reassociation is because of an intrinsic or extrinsic defect in the central macrophage, we generated reciprocal BM chimeras using CD45.1 congenic Sle1.Yaa mice. 8-wk-old huTLR8tg.Sle1.Yaa-recipient mice were unable to support BM transfers and failed to reconstitute after irradiation. Chimeras made using either 30% or 50% CD45.2.huTLR8tg.Sle1.Yaa and 70% or 50% CD45.1.Sle1.Yaa BM into CD45.1.Sle1.Yaa recipients reconstituted (Fig S16A and B), but none became anemic over >6 mo of observation. In these mice, all myeloid cell subsets from the huTLR8tg.Sle1.Yaa donors had a substantial growth advantage, suggesting an intrinsic exaggerated response of myeloid cells to TLR8 ligation (Fig 7H and I).

# Discussion

Erythropoiesis generates a large load of nuclear material that must be disposed of in a noninflammatory fashion. Genetic defects of the innate immune system that prevent safe disposal of nucleic acids have been linked to anemia ranging in severity from lethal anemia in utero (Kawane et al, 2001; Postel et al, 2009; Liddicoat et al, 2015, 2016) to dysfunctional erythropoiesis with normochromic, normocytic anemia (Rego et al, 2018).

We report here a new model of acquired fatal anemia in lupus-prone mice caused by the presence of functional human TLR8. This syndrome requires the presence of the *Sle1* and *Yaa* loci: the *Sle1* locus confers loss of tolerance to nucleic acids, whereas the *Yaa* locus in males confers an extra copy of *Tlr7* that drives pathogenic autoantibody formation (Soni et al, 2014). Male Sle1.Yaa mice develop lupus nephritis and mild-to-moderate anemia and die by 1 yr of age. The early onset of lethal anemia in huTLR8tg Sle1.Yaa mice occurs in a stochastic fashion shortly after high titer autoantibodies appear in the serum. Anemia is associated with increased formation of trabecular bone, like that described in *DNAse2a*-deficient mice in which bone accrual requires both STING and the endosomal TLR transporter Unc93b1 (Baum et al, 2017; Marshak-Rothstein et al, 2020). *DNAse2a* is highly expressed by EMBI macrophages (Li et al, 2019) and our data suggest that loss of *DNAse2a* in cluster Mφ2 could contribute to the bone phenotype huTLR8tg.Sle1.Yaa mice.

Acquired disturbance of the BM erythropoietic niche in huTLR8tg.Sle1.Yaa mice is associated with loss of BM EMBIs, specialized structures in which RBC precursors attach to a central macrophage that provides erythroblasts with nutrients and proliferation and survival signals, and phagocytoses the pyrenocyte in a noninflammatory manner (Manwani & Bieker, 2008; Hom et al, 2015). Our results demonstrate a block in the differentiation of erythroid precursors at the CFU-E to ProEB transition in the BM of huTLR8tg.Sle1.Yaa mice. Differentiation

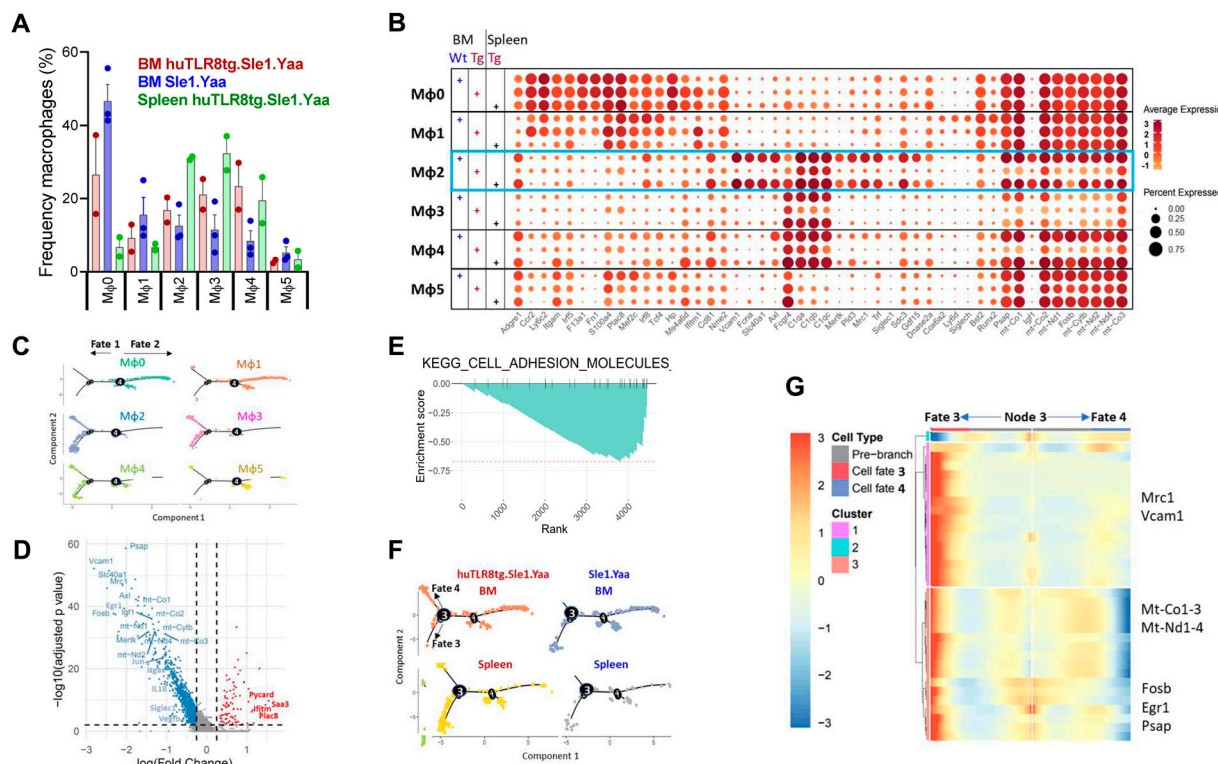

**Figure 6. Characterization of central macrophage subpopulations.**
**(A)** Frequency of cells in each cluster among cells belonging to the macrophage lineage. **(B)** Bubble plot shows genes that differentiate the macrophage clusters and their expression in huTLR8tg.Sle1.Yaa (Tg) BM and spleen compared with Sle1.Yaa (Wt) BM. **(C)** Trajectory analysis of clusters Mφ0 to Mφ5 shows two trajectories of cells diverging from Node 4 with clusters Mφ0 and Mφ1 separating from the other clusters (see Fig S13B). **(D)** Volcano plot shows differentially expressed genes in cluster Mφ2 from huTLR8tg.Sle1.Yaa BM compared with Sle1.Yaa BM identified by KEGG analysis. **(E)** GSEA analysis of the KEGG cell adhesion pathway (See Fig S10B). **(F, G)** Trajectory analysis shows two trajectories of cells within the Mφ2–4 clusters diverging from Node 3 with Fate 3 being predominant in Sle1.Yaa BM and huTLR8tg.Sle1.Yaa spleens and Fate 4 being predominant in huTLR8tg.Sle1.Yaa BM.

of these early erythroid precursors requires down-regulation of glycolysis and an increase in oxidative phosphorylation (Isern et al, 2011; Richard et al, 2019). This switch is attenuated in CFU-E of the transgenic BMs but not in splenic CFU-E, suggesting that its loss is not because of an intrinsic RBC defect.

Analysis of BM chimeras clearly showed an intrinsic hyper-responsiveness of the huTLR8tg myeloid cell compartment to growth signals. However, when we transferred huTLR8tg.Sle1.Yaa BM to Sle1.Yaa mice, the chimeric mice did not become anemic even after more than 6 mo of follow-up. Thus, they did not acquire the BM phenotype of nonirradiated mice. It remains unclear whether this is because of the protection by the wild type BM, differences in the WT hematopoietic niche or a result of irradiation. We were not able to reproduce the defect in huTLR8tg.Sle1.Yaa recipients as they could not reconstitute transferred BM.

Our single-cell studies define the heterogeneity of macrophages within the EMBIs of Sle1.Yaa mice and define two major subsets, one expressing *Ly6c2* and *Ccr2* and one expressing *Cd81* and *C1q*. Within the CD81/C1q subset, we show that cluster Mφ2 uniquely expresses the canonical phenotypic markers of central macrophages. Within BM EMBIs of transgenic mice, this cell cluster acquires a pro-inflammatory profile and down-regulates genes that are required for adhesion and provision of nutrients to erythroid precursors (Chasis &

Mohandas, 2008). Of the down-regulated genes, *Itgav, Axl, Mertk, Dnase2a,* and *cMaf* deficiencies are associated with impaired erythropoiesis (Li et al, 2020), strongly suggesting that the alteration in gene programs in central macrophages of huTLR8tg.-Sle1.Yaa mice causes the inability of CFU-E to proliferate, switch their metabolism, and differentiate to the ProEB stage. Ly6C/CCR2-expressing cells in the EMBIs are likely myeloid precursors that differentiate in parallel with erythroid cells (Romano et al, 2022).

The defect in BM erythropoiesis in huTLR8tg.Sle1.Yaa mice induces extensive stress erythropoiesis. In contrast to BM EMBIs, islands in the spleen remain capable of promoting erythroblast maturation in the transgenic mice. Ly6C/CCR2-expressing macrophages are observed in the spleens of mice with acute anemia and stress erythropoiesis, but then lose the expression of these cell surface markers as they mature (Liao et al, 2018).

Why erythroid niches collapse in the BM, whereas they expand in the spleen is still unknown. Although both BM and spleen stress BFU-E need Epo to differentiate, they are regulated differently and derive from different progenitors. BFU-E in the spleen depend on Hedgehog ligands and BMP4 that are not available in the BM and high levels of Gdf15 that regulates cell metabolism. In the setting of inflammation, the transcription factor SPI-C promotes the expression of *Gdf15* and *Bmp4* and *enhances stress erythropoiesis in*

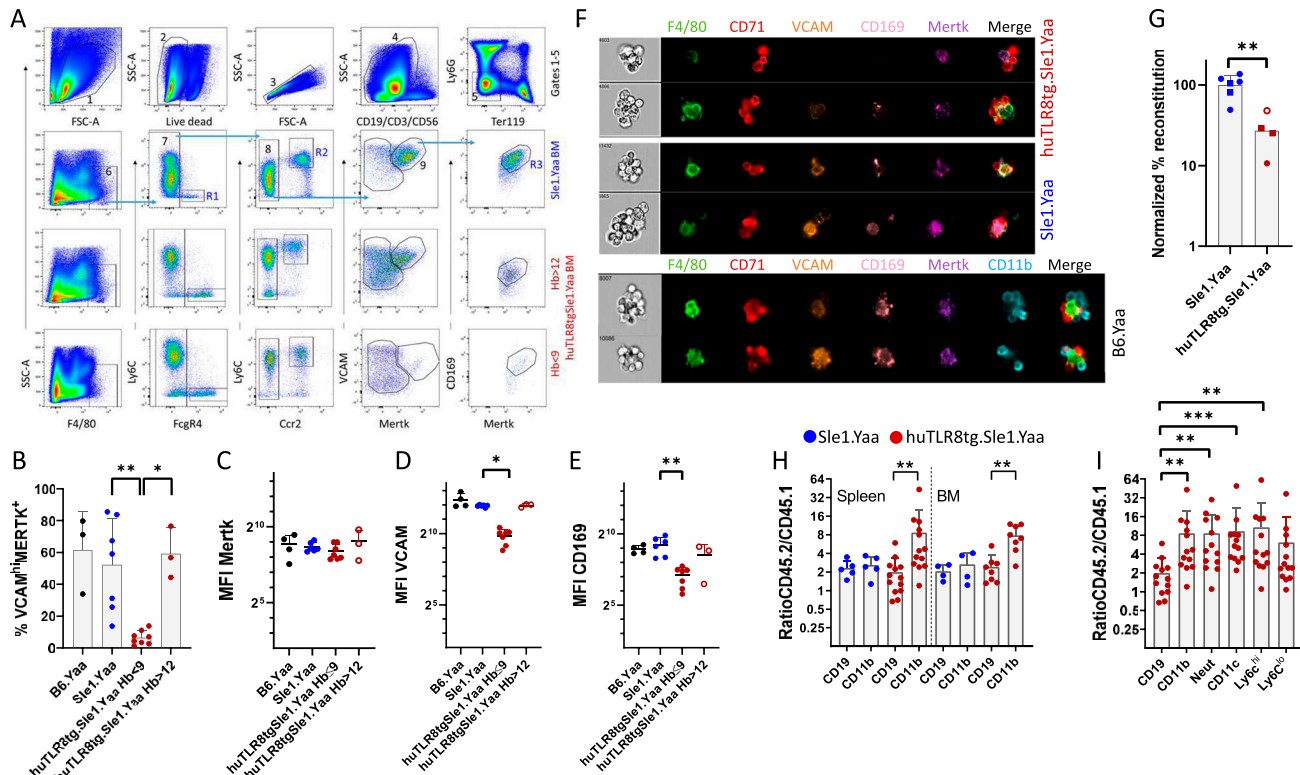

**Figure 7. Flow cytometry of BM and spleen central macrophages.**
**(A)** Gating strategy for BM central macrophages. Whole BM cells were gated on live-singlet CD19⁻/CD3⁻/CD56⁻/Ly6G⁻/Ter119⁻ cells (upper panel), then for F4/80ʰⁱ cells and sequentially for FcYR4 (gate R1), Ly6C/CCR2 (gate R2), and VCAM/Mertk/CD169 (gate R3). Note the loss of VCAMʰⁱ/Mertkʰⁱ cells in the severely anemic huTLR8tg.Sle1.Yaa mouse (bottom panel) compared with Sle1.Yaa and non-anemic huTLR8tg.Sle1.Yaa controls (middle panels). **(A, B)** Quantification of VCAMʰⁱ/Mertkʰⁱ cells in BMs of the indicated strains gated as a percentage of F4/80ʰⁱ cells as established in (A). **(A, C, D, E)** Median fluorescence intensity of MERTK (C), CD169 (D), and VCAM (E) in Gate 9 as established in (A). **(B, C, D, E)** Kruskal–Wallis ANOVA followed by Dunn's multiple comparisons test. **(F)** Representative ImageStream analysis of EMBIs isolated from the BM of indicated strains confirms the loss of VCAM and CD169 from the central macrophages of the anemic huTLR8tg.Sle1.Yaa mouse compared with Sle1.Yaa and B6.Yaa controls. **(G)** Reassociation frequencies of EMBIs from whole BMs (circles) or from isolated BM EMBIs (squares) from huTLR8tg.Sle1.Yaa with reticulocyte count >7% and age-matched Sle1.Yaa mice. Reassociation frequency was adjusted to the mean of the Sle1.Yaa controls for each experiment given a value of 100%. Closed red symbols indicate huTLR8tg.Sle1.Yaa mice with Hb < 9. Mann–Whitney t test **P < 0.01. **(H, I)** Flow cytometry of spleen (H, I) and BM (H) from BM chimeras according to the gating strategy in Fig S16 shows the ratio of CD45.2 (huTLR8tg.Sle1.Yaa or Sle1.Yaa) to CD45.1 (Sle1.Yaa) B cells and myeloid cells (H) and myeloid subsets (I). **(B, C, D, E, H, I)** *P < 0.05; **P < 0.01, ***P < 0.001. Each data point represents a single mouse.

*the spleen* (Bennett et al, 2019). Our single-cell data confirm that spleen Mφ2 macrophages from huTLR8tg.Sle1.Yaa mice express higher levels of *SpiC* and *Gdf15* than their BM counterparts.

The effect of innate immune stimulation on BM erythropoiesis has previously been studied in nonautoimmune mice in which LPS suppresses erythropoiesis at the CFU-E/ProEB stage, while enhancing stress erythropoiesis through induction of SpiC. Similar findings have been observed in response to exogenous administration of G-CSF and Flt3 ligands (Bisht et al, 2020; Tay et al, 2020; Lévesque et al, 2021). Flt3 ligand is known to be up-regulated in the BMs of Yaa mice (Adachi et al, 2002) and T cells expressing membrane Flt3 ligand have been reported in the BMs of lupus patients with aplastic anemia (Pfister et al, 2000). In the case of LPS, collapse of BM EMBIs is MyD88-dependent but independent of inflammatory cytokines including TNF, IL1 and G-CSF (Bisht et al, 2020; Lévesque et al, 2021); these findings are consistent with the known signaling of huTLR8 through MyD88 and our own unpublished observation that anemia is TNF-independent in our model. These conditions differ in mechanism from the anemia of inflammation

that is associated with a cytokine-mediated decrease in BM erythropoiesis in favor of innate immune cell (myeloid cell) production (Dulmovits et al, 2022).

In our model, loss of BM erythropoiesis is progressive and irreversible. Although stress erythropoiesis can compensate for acute disturbances of BM erythropoiesis, it has previously been suggested that there may be a limited capacity for stress erythropoiesis under conditions of chronic blood loss or chronic inflammation (Paulson et al, 2020). Indeed, repeated cycles of stress erythropoiesis because of sustained inflammation may eventually result in attenuation of the stress response (Bennett et al, 2019). We show here that stress erythropoiesis continues throughout the disease course but increased hemophagocytosis because of the expansion of several subsets of hemophagocytic macrophages (Akilesh et al, 2019) eventually exceeds the capacity of stress erythropoiesis to replace the red cell mass. Hemophagocytosis has been reported in SLE patients and is associated with flare or infections, both of which are expected to confer an increased load of TLR ligands (Lambotte et al, 2006; Kim et al, 2012). Although

uncommon, this is a severe manifestation of lupus that is reversed by immunosuppression. Importantly, we did not detect an intrinsic RBC defect that made them more prone to hemophagocytosis.

In summary, we present evidence that excess innate stimulation through the endosomal RNA sensor huTLR8 alters the ability of EMBI macrophages to support erythropoiesis within the BM niche, likely through a mechanism involving MyD88. This is followed by stress erythropoiesis and inflammatory hemophagocytosis. Because anemia only occurs when both the Sle1 and *Yaa* loci are present, the observed phenotype requires the presence of pathogenic autoantibodies and/or the provision of an excess load of huTLR8 ligands, explaining why it is acquired in a stochastic fashion, like what is observed in patients with SLE. BMs of SLE patients are abnormal with dyserythopoiesis, hemophagocytosis, and stromal damage/disorganization among other abnormalities (Zhuang et al, 2014; Anderson et al, 2018). Further analysis of our model may help in understanding the pathogenesis of chronic anemia in patients with SLE and whether modulating endosomal TLRs, particularly TLR8, can treat the chronic anemia of SLE.

# Materials and Methods

## Mice

C57BL/6 huTLR8 BAC transgenic (Clone 8) mice (Guiducci et al, 2013), a gift from Dr. Christiana Guiducci (Dynavax), were crossed to C57BL/6.Yaa (Jackson Laboratories) and Sle1 mice. C57BL/6.Yaa mice (B6.Yaa) and huTLR8.B6.Yaa (B6.Yaa tg) mice behaved similarly and were used as negative controls. Mice were typed for Sle1 using three primer sets (https://www.jax.org/strain/021569) and for hetero- or homozygosity of the huTLR8 transgene using droplet digital DNA PCR (Fig S17A–C). The transgene was subsequently mapped using targeted sequencing by proximity ligation (Cergentis) to a single complex integration site on chromosome 10 (Fig S17D). This information allowed us to design a commercial typing assay for the transgenic and WT alleles (Transnetyx). The two assays were 100% concordant.

## BMDM stimulation and metabolic assays

BM cells from B6.huTLR8tg mice or C57BL/6 littermates were plated in triplicates in 96-well plates stimulated with TLR7-specific agonist CL264 or TLR8-specific agonist TL-506 (InvivoGen) at the indicated concentrations for 24 h and then TNFα from the supernatants was measured by ELISA (R and D). $2 \times 10^5$ BMDM from TLR7-deficient mice were plated overnight in XF media on Seahorse culture plates that were pre-coated with Cell-Tak (Corning cell and tissue adhesive) for 20 min at RT, washed with $H_2O$ and air dried before cell plating. Seahorse culture plates were pre-coated with Cell-Tak (25 $\mu l$ of 11.2 μg/mL) for 20 min at RT (Corning cell and tissue adhesive), washed with $H_2O$, and air dried before cell plating. Cells were stimulated with agonists CL075 (3M002; InvivoGen) or TL-506 (InvivoGen) for 24 h. BMDM extracellular acidification rate was then measured using a Seahorse XF96 Extracellular Flux Analyzer and the Glycolysis Stress Test kit according to manufacturer's instructions (Agilent). Data were normalized to the cell number.

## Autoantibodies

Mice were bled every 2–4 wk and urine was tested for proteinuria by dipstick (Multistick; Thermo Fisher Scientific) (Mihara et al, 2000). Antibodies to chromatin, Sm/RNP, and cardiolipin/β2 glycoprotein1 were measured using ELISA (Akkerman et al, 2004; Ramanujam et al, 2006). A high-titer serum was run in serial dilutions on each plate as a standard. OD values were converted to units using regression analysis (Prism Version 9).

## Complete blood counts

Hematologic parameters were obtained using an ADVIA2120i (Siemens).

## Flow cytometry

Single-cell suspensions from spleens and BMs were stained in 2.5% FBS/0.05% $NaN_3$ in PBS containing fluorochrome-conjugated antibody panels listed in Table S2. Samples were acquired on an LSR Fortessa or a BD Symphony flow cytometer (BD Biosciences) and analyzed with FlowJo v10 software (Tree Star).

## Flow sorting and huTLR8 qRT-PCR

Single-cell suspensions from spleens were stained in 2.5% FBS/0.05% $NaN_3$ in PBS containing fluorochrome-conjugated antibodies to CD19, CD3, Ly6G, CD11b, and CD11c and live singlet B cells (CD19⁺), T cells (CD3⁺), macrophages (CD11b⁺/CD11c^lo), and dendritic cells (CD11b⁺/CD11c^hi) were flow sorted. BMs were enriched for CD45⁺ cells using EasySep Mouse CD45 Positive Selection Kit and then stained with fluorochrome-conjugated antibodies to CD19, CD3, Ly6G, Ly6C, VCAM1, and F4/80. F4/80^hi/CD3⁻/CD19⁻/Ly6G⁻ cells including Ly6C^hi and VCAM^hi cells were flow sorted. The CD45-depleted portion of the BM was then enriched for CD117⁺ cells using EasySep Mouse Hematopoietic Progenitor Cell Isolation Kit. Cells were stained with fluorochrome-conjugated antibodies to CD45, Ter119, CD71, cKit, Sca-1, and CD55 and CD45⁻/cKit⁺/Sca-1⁻/CD55⁺ erythroid progenitors were flow sorted. Untouched neutrophils were obtained from the peripheral blood after magnetic bead enrichment (Miltenyi Biotec). Antibodies are described in Table S2. Samples were subjected to qRT-PCR using intron-spanning huTLR8 primers (Forward AGAGGGTACCATTCTGCGCT; Reverse GCAGGTCAG-CATTGACGACT) and primers for βactin (B-ACTIN-F: ACCGTGAAAA-GATGACCCAG; B-ACTIN-R: AGCCTGGATGGCTACGTACA).

## Pathology

Hematoxylin & Eosin staining and TLR8 staining (#LS C161923; LSBio) of FFPE tissue were performed by a commercial vendor (Histowiz). Touch preps of freshly cut spleens were dried and stained using Wright Giemsa stain.

## Immunohistochemistry

Spleens were fixed in cold 3% PFA 45 min, washed with PBS, and incubated in 30% sucrose solution for 2 h before being embedded

in OCT. 10 micron frozen sections of the spleens were blocked with 10% NGS and stained with four color panels using combinations of antibodies to F4/80, B220, CD11b, and Ter119 as described in Table S2. Images were obtained using a Zeiss Apotome confocal microscope.

## Red cell survival

For biotinylation of erythrocytes, 1 mg of EZ Link Sulfo-NHS biotin (Thermo Fisher Scientific) was dissolved in sterile saline and administered intravenously. Biotinylated erythrocytes were quantified by flow cytometry using Streptavidin-PE. CFSE labeling and transfusions were performed as previously described (Mock et al, 2014; Donnenberg et al, 2019). Spleens were harvested after 24 h, dissociated using the gentleMACS Dissociator and spleen dissociation kit (Miltenyi Biotec), and stained with antibodies to myeloid cell subsets as shown in Table S2. Cytospins of stained cells were examined using confocal microscopy.

## EMBI preparation

BM cells were flushed out of tibiae, femurs, and pelvic bones with a 26Gx3/8 1 mL syringe (BD Safetyglide) containing IMDM (Gibco) supplemented with 20% FBS (Hyclone), 1 mM CaCl2, and 1 mM MgCl2 (EMBI Buffer) at RT. BM large aggregates were gently homogenized with a 1 mL pipette, filtered through a 70 $\mu$m cell strainer, and resuspended into a 3-mL EMBI buffer. EMBI buffer supplemented with either 3% or 1.5% BSA wt/vol (Protease-free powder; Thermo Fisher Scientific) was prepared at RT. Using a 50-mL conical tube, a discontinuous gradient was prepared with 5 mL fractions of 3%, 1.5%, and 0% EMBI buffers starting with the 3% fraction in the bottom. The 3 mL cell suspension in the EMBI buffer was gently transferred onto the gradient and incubated for 20 min at RT for sedimentation. The top layers were discarded and the 3% fraction containing EMBIs was collected. EMBIs were counted using a glass chamber.

For single-cell RNA sequencing, the fraction containing EMBIs was centrifuged at 4°C for 5 min at 400$g$. 10 mL pre-chilled PBS without calcium–magnesium supplemented with 0.5% BSA and 5 mM EDTA were added to the pellet and the suspension was mixed for 30 min at 4°C on a tube rotator. The single-cell suspension was then centrifuged (5 min, 400$g$, 4°C), washed with 1 mL prechilled PBS 0.04% BSA, and filtered through a 40 $\mu$m cell strainer (Bel-Art Products). Cell concentration was adjusted following the manufacturer's instructions (https://www.10xgenomics.com/support/single-cell-gene-expression/documentation/steps/sample-prep/single-cell-protocols-cell-preparation-guide).

## Reassociation of EMBIs

BM cells were flushed out of tibiae, femurs, and pelvic bones with a 26Gx3/8 1 mL syringe (BD Safetyglide) containing IMDM (Gibco) supplemented with 20% FBS (Hyclone) and 2 mM EDTA and either used to isolate EMBIs as above or used without further isolation. Cell clusters were dissociated by multiple gentle passages through the same 26Gx3/8 1 mL syringe until a single-cell suspension was obtained. This was confirmed by bright field microscopy imaging

(M7000 Imaging System; EVOS) using a diluted aliquot in one well of a 96-well plate. Once a single-cell suspension was obtained, the cells were passed through a 70 $\mu$m cell strainer, washed, and resuspended in IMDM supplemented with 20% FBS 1 mM CaCl2 and 1 mM MgCl2 (EMBI Buffer) at RT. Cells were counted using a Countess 3 FL Automated Cell Counter and resuspended at $40 \times 10^6$/mL. For reassociation, 1 mL of single cell suspension was incubated at 37°C for 2 h with gentle rocking in one well of a 12-well plate. Reconstituted EMBIs were passed through a 70 $\mu$m cell strainer and enriched using a BSA discontinuous gradient as above. The 3% BSA EMBI fraction was collected, counted, and used for imaging experiments. Reassociation frequency was adjusted to the mean of the Sle1.Yaa controls for each experiment given a value of 100%.

## Imaging flow cytometry of EMBIs

PFA (EM grade, Electron Microscopy Sciences) was added to the 3% BSA fraction to a final concentration of 4% and the suspension was transferred to a 5 mL conical tube. The EMBIs were incubated for 20 min on a rocker with gentle shaking at RT. EMBIs were centrifuged at 400$g$ 5 min, RT, and washed twice with 1 mL PBS. Cells were resuspended in 100 $\mu$l PBS 0.5% BSA containing anti-CD16/32 and stained for 30 min at RT with antibodies to F4/80, CD11b, CD71, Vcam1, CD169, and MERTK (Table S2). After washing with PBS 0.5% BSA, data were acquired on Amnis ImageStreamX Mk II Imaging Flow Cytometer and analyzed with IDEAS 6.2 software.

## BM chimeras

CD45.1 Sle1.Yaa and CD45.2 huTLR8tg.Sle1.Yaa male mice were lethally irradiated and male recipients of both genotypes were reconstituted with mixed BM from CD45.2 huTLR8tg.Sle1.Yaa and CD45.1 Sle1.Yaa donors in a 1:1 or 1:3 ratio. Recipients were monitored for anemia for more than 6 mo before euthanization and flow cytometry of BMs and spleens.

## 10X Genomics chromium single-cell RNA sequencing

Single cells were obtained from dissociated EMBIs from BMs and spleens of two severely anemic huTLR8tg mice and three Sle1.Yaa non-transgenic mice aged 100–110 d. 8,000 cells from each sample were subjected to scRNAseq using 10X Genomics 3′ kit v3 following the manufacturer's instructions. Raw data are available in GEO–accession number GSE189532.

## Single-cell RNA sequencing data analysis

Raw sequencing reads were aligned, counted, and demultiplexed by cellranger v5.0.0 with default parameters using mouse genome mm10 as reference. The raw gene-cell counts were then used as input for Seurat 3.1.5 for quality control, normalization, unsupervised clustering, and visualization. Genes expressed in less than three cells and cells expressed less than 200 genes were filtered out as poor quality. Cells with >= 20% RNA content from mitochondrial genes were considered dead and filtered out. Doublets were predicted using scrublet v0.2.3 (Wolock et al,

2019) and cells with doublet score >0.1 were removed from further analysis. Raw read counts were normalized by read depth and log transformed with NormalizeData using scale factor 10,000. The normalized expression values for each gene were further centered using ScaleData for downstream clustering analysis. Raw data are available in GEO under accession number GSE189532.

Cells from different samples were integrated using Harmony 1.0 (Korsunsky et al, 2019) with the RunHarmony method implemented in Seurat. Cells were then clustered with Seurat in an unsupervised manner using FindNeighbors with the first 15 harmony dimensions and FindClusters with resolution 0.8. UMAPs were generated with RunUMAP with the first 15 harmony dimensions. Markers of each cell type were identified with FindAllMarkers with the default testing method, Wilcoxon rank sum test. Differentially expressed genes (DEGs) were identified using FindMarkers by comparing cells between groups within each cell type with the default testing method, Wilcoxon rank sum test. Genes with $P$-value <= 0.01 and log fold change >= 0.25 or <= −0.25 were identified as significant DEGs. After removal of five small contaminating clusters of T cells, B cells, and platelets comprising ~15% of total cells, we analyzed 19,309 BM and 4,794 spleen EMBI cells.

The trajectory trees for clusters E1 and Mφ2 were constructed separately with Monocle 2 (Qiu et al, 2017) using the top 1,000 DEGs between cell types. Branch-dependent genes of particular nodes were identified using BEAM function.

## Statistics

Results represent the mean ± SD. Statistical analyses are described in the figure legends and were performed using GraphPad Prism Version 9 software using unpaired nonparametric $t$ test (Mann–Whitney) or Kruskal–Wallis ANOVA with Dunn's corrections for multiple comparisons for >2 groups. A $P$-value less than 0.05 was considered significant.

## Data Availability

Raw data are available in GEO under accession number GSE189532.

## Supplementary Information

## Acknowledgements

We acknowledge technical assistance of Shani Martinez, Catherine Diadhiou, Haiou Tao, and Ke Lin. This work was supported by the Lupus Research Alliance (to A Davidson), the NIH HL152099 (to L Blanc and T Kalfa), HL144436 (to L Blanc), HL146528, and USDA NIFA Hatch Project PEN04771 accession #0000005 (to R Paulson).

## Author Contributions

NI Maria: conceptualization, data curation, formal analysis, investigation, methodology, and writing—original draft.
J Papoin: conceptualization, data curation, formal analysis, investigation, methodology, and writing—original draft.
C Raparia: formal analysis, investigation, and methodology.
Z Sun: data curation, formal analysis, and writing—original draft.
R Josselsohn: data curation, formal analysis, and investigation.
A Lu: formal analysis and methodology.
H Katerji: formal analysis and methodology.
MM Syeda: data curation, formal analysis, and methodology.
D Polsky: conceptualization, data curation, and methodology.
R Paulson: methodology and writing—review and editing.
T Kalfa: conceptualization and methodology.
BJ Barnes: resources, methodology, and writing—review and editing.
W Zhang: data curation, formal analysis, supervision, and writing—original draft.
L Blanc: conceptualization, resources, data curation, formal analysis, supervision, methodology, and writing—original draft, review, and editing.
A Davidson: conceptualization, data curation, formal analysis, supervision, funding acquisition, investigation, methodology, project administration, and writing—original draft, review, and editing.

## Conflict of Interest Statement

The authors declare that they have no conflict of interest.

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
