## [Reviewer comments · Life Science Alliance]

Life Science Alliance

Human TLR8 induces inflammatory bone marrow erythromyeloblastic islands and anemia in SLE-prone mice

Naomi Maria, Julien Papoin, Chirag Raparia, Zeguo Sun, Rachel Josselsohn, Ailing Lu, Hani Katerji, Mahrukh Syeda, David Polsky, Robert Paulson, Theodosia Kalfa, Betsy Barnes, Weijia Zhang, Lionel Blanc, and Anne Davidson

DOI: <https://doi.org/10.26508/lsa.202302241>

Corresponding author(s): Anne Davidson, Feinstein Institutes for Medical Research; Anne Davidson, Feinstein Institutes for Medical Research; and Lionel Blanc, Feinstein Institute for Medical Research

Review Timeline:

Submission Date:	2023-06-27
Editorial Decision:	2023-06-29
Revision Received:	2023-06-29
Editorial Decision:	2023-07-04
Revision Received:	2023-07-04
Accepted:	2023-07-06

Transaction Report:

Please note that the manuscript was previously reviewed at another journal and the reports were taken into account in the decision-making process at *Life Science Alliance*. Since the original reviews are not subject to Life Science Alliance's transparent review process policy, the reports and author response cannot be published.

June 29, 2023

Re: Life Science Alliance manuscript #LSA-2023-02241-T

Dr. Anne Davidson
Feinstein Institutes for Medical Research
Institute of Molecular Medicine
350 Community Drive
Manhasset, New York 11030

Dear Dr. Davidson,

Thank you for submitting your manuscript entitled "Human TLR8 induces inflammatory erythromyeloblastic islands the bone marrow leading to fatal anemia in SLE-prone mice" to Life Science Alliance. We invite you to submit a revised manuscript addressing the following Reviewer points:

- Add discussion related to Reviewer 4's points #3 and 4.

Thank you for this interesting contribution to Life Science Alliance. We are looking forward to receiving your revised manuscript.

Sincerely,

Eric Sawey, PhD
Executive Editor
Life Science Alliance
<http://www.lsa-journal.org>

B. MANUSCRIPT ORGANIZATION AND FORMATTING:

July 4, 2023

RE: Life Science Alliance Manuscript #LSA-2023-02241-TR

Dr. Anne Davidson
Feinstein Institutes for Medical Research
Institute of Molecular Medicine
350 Community Drive
Manhasset, New York 11030

Dear Dr. Davidson,

Thank you for submitting your revised manuscript entitled "Human TLR8 induces inflammatory bone marrow erythromyeloblastic islands and anemia in SLE-prone mice". We would be happy to publish your paper in Life Science Alliance pending final revisions necessary to meet our formatting guidelines.

- please upload your Tables in editable .doc or Excel format
- please upload all figure files as individual ones, including the supplementary figure files; all figure legends should only appear in the main manuscript file
- please add your main and supplementary figure legends to the main manuscript text after the references section
- please note that titles in the system and on the manuscript file must match
- please remove a number of words and key points from the manuscript text
- please consult our manuscript preparation guidelines <https://www.life-science-alliance.org/manuscript-prep> and make sure your manuscript sections are in the correct order
- please use the [10 author names et al.] format in your references (i.e., limit the author names to the first 10)
- we encourage you to revise the figure legends for Figure 3 such that the figure panels are introduced in an alphabetical order
- please add callouts for Figures 5E, S8A-C; S10A-C; S15C; S16A-D; S17A,B to your main manuscript text
- supplementary methods and references need to be incorporated into the main text

Figure checks:

- please revise the inset position in Figure S1B so that they match the zoomed-in parts;
- please add scale bars in Figure S2

A. FINAL FILES:

- An editable version of the final text (.DOC or .DOCX) is needed for copyediting (no PDFs).
- High-resolution figure, supplementary figure and video files uploaded as individual files: See our detailed guidelines for preparing your production-ready images, <https://www.life-science-alliance.org/authors>
- Summary blurb (enter in submission system): A short text summarizing in a single sentence the study (max. 200 characters including spaces). This text is used in conjunction with the titles of papers, hence should be informative and complementary to

the title. It should describe the context and significance of the findings for a general readership; it should be written in the present tense and refer to the work in the third person. Author names should not be mentioned.

B. MANUSCRIPT ORGANIZATION AND FORMATTING:

Sincerely,

July 6, 2023

RE: Life Science Alliance Manuscript #LSA-2023-02241-TRR

Dr. Anne Davidson
Feinstein Institutes for Medical Research
Institute of Molecular Medicine
350 Community Drive
Manhasset, New York 11030

Dear Dr. Davidson,

Thank you for submitting your Research Article entitled "Human TLR8 induces inflammatory bone marrow erythromyeloblastic islands and anemia in SLE-prone mice". It is a pleasure to let you know that your manuscript is now accepted for publication in Life Science Alliance. Congratulations on this interesting work.

DISTRIBUTION OF MATERIALS:

Again, congratulations on a very nice paper. I hope you found the review process to be constructive and are pleased with how the manuscript was handled editorially. We look forward to future exciting submissions from your lab.

Sincerely,
